# Thriving at Work as a Mediator of the Relationship between Transformational Leadership and Innovative Work Behavior

**Nada Alwahhabi [1], Suad Dukhaykh [1,*] and Wadi B. Alonazi [2]**

[1] Management Department, College of Business Administration, King Saud University, Riyadh 11451, Saudi Arabia; 442204228@student.ksu.edu.sa

[2] Health Administration Department, College of Business Administration, King Saud University, Riyadh 11587, Saudi Arabia; waalonazi@ksu.edu.sa

* Correspondence: sdukhaykh@ksu.edu.sa

**Abstract:** Thriving at work is a positive psychological state in which individuals experience a sense of vitality and learning. The purpose of this quantitative study is to examine the effect of transformational leadership on innovative work behavior and the mediating effect of thriving at work on the relationship between transformational leadership and innovative work behavior among private sector employees in the Kingdom of Saudi Arabia. A cross-sectional research design was used to collect data from 224 private sector employees. The results of the study reveal that transformational leadership is positively related to thriving at work and thriving at work is positively related to innovative work behavior. Furthermore, thriving at work fully mediates the relationship between transformational leadership and innovative work behavior. The theoretical and practical implications are discussed.

**Keywords:** transformational leadership; thriving at work; innovative work behavior; Saudi Arabia

## 1. Introduction

Human resources are essential for organizations' sustainability, in addition to the economic and environmental aspects as the triple bottom line of organizational sustainability [1]. However, the human aspect of sustainability has received considerably less attention than the environmental and economic aspects [2]. According to Spreitzer, Porath [2], thriving at work is an essential mechanism for comprehending the human aspect of sustainability. Thriving at work is a growing concept that has gained considerable attention in the field of organizational behavior as a component of human sustainability and long-term performance [3]. Thriving was defined as a psychological state when individuals feel both alive and like they were learning at work. [4]. It is a desirable self-regulatory subjective state with two dimensions: vitality (affective) and learning (cognitive) [4]. Vitality refers to the feeling of being alive, energized, enthusiastic, and excited at work [5]. Learning is about developing through gaining knowledge, skills, and other qualities [4]. Thriving is widely believed to have an important role in improving both short-term individual performance and long-term workplace adaptability [4]. For instance, thriving at work has been proven to be related to a variety of essential organizational outcomes such as job satisfaction, self-development, organizational commitment, organizational citizenship behavior, and inventive work behavior [6–10].

The main model of thriving at work was called the socially embedded model of thriving and it was developed by Spreitzer, Sutcliffe [4]. This model explained how certain contextual features such as decision-making autonomy and a climate of respect and trust may enable employees' agentic behaviors such as task focus and exploration. These behaviors, as well as the additional job resources, promoted employees thriving at work. Further, thriving leads to positive outcomes for employees such as in their development and health [4].

The literature on employees' thriving at work has significantly expanded since Spritzer and her colleagues established the socially embedded model of thriving [11]. However, literature on thriving at work is still scarce [3,12]. For instance, Abid and Contreras [12] did a meta-analysis and found that most studies on thriving at work were done in developed countries like the US, China, Australia, Belgium, and France. Abid and Contreras [12] said that researchers should add more regional studies from emerging economies to make management and business studies more useful, in-depth, and broad. In addition, previous studies highlighted that, in thriving studies, transformational leadership needed to be investigated in different contexts [13,14]. Furthermore, although several studies have examined the impact of transformational leadership on innovative work behavior [15,16], to the best of the researchers' knowledge, limited studies have been conducted to understand the mediating effect of thriving at work in the relationship between transformational leadership and innovative work behavior.

For this reason, based on the socially embedded model of thriving [4], we proposed that transformational leadership positively influences employees' thriving at work, which leads to innovative work behavior. Transformational leadership is described as leaders who inspire followers' ambitions for success and self-improvement and support the growth of groups and organizations [17]. Previous studies agreed that transformational leadership develops followers to a higher level by meeting their basic needs [18]. In addition, transformational leadership behaviors have been shown to promote a wide range of positive outcomes, including task performance, citizenship behavior, work satisfaction, follower motivation, and employee innovation [18–20].

In today's competitive world, organizations constantly seek a sustainable competitive advantage. De Jong and Den Hartog [21] stated that innovation is essential for organizations in the private sector that are seeking a sustainable competitive advantage. Innovation is commonly agreed to be crucial for an organization's effectiveness [22]. An organization's innovation depends on its employees' inventive work behavior [21]. Almost 80% of creative ideas in the workplace are sourced from the innovative behavior of employees [23]. This behavior involves the creation of new ideas, technology, and methods, as well as the trial and utilization of new techniques related to business processes, mainly work aspects [15].

Therefore, the aim of this study is to examine the effect of transformational leadership on employees' thriving at work and the mediating effect of thriving at work on the relationship between transformational leadership and innovative work behavior among private sector employees in the Kingdom of Saudi Arabia. This aim will be addressed by answering the following question. What is the effect of transformational leadership on employees' thriving at work, and how does this relationship impact employees' innovative work behavior in the private sector of the Kingdom of Saudi Arabia?

The present study contributes to the literature on employees' thriving at work by investigating the impact of transformational leadership on employees' thriving at work and innovative work behavior in the private sector of the Kingdom of Saudi Arabia. This study also contributes to the body of knowledge by incorporating regional studies from an emerging economy, specifically the private sector of the Kingdom of Saudi Arabia. In addition, this study provides evidence for the mediating effect of thriving at work in the relationship between transformational leadership and innovative work behavior. Finally, this study's findings highlight the importance of transformational leadership in promoting employees' thriving at work and innovative work behavior and have practical implications for managers and leaders in the private sector of the Kingdom of Saudi Arabia.

The rest of this study is organized as follows: The following section is a literature review and builds the hypotheses; it is followed by the study methods and results, which are followed by a discussion. Finally, theoretical, and managerial implications are discussed at the end of this study.

## 2. Literature Review and Hypotheses Development

### 2.1. Linking Transformational Leadership and Innovative Work Behavior

Transformational leadership style is described as leaders who inspire followers' ambitions for success and self-improvement and support the growth of groups and organizations [17]. After conducting a literature review on transformational leadership, Podsakoff, MacKenzie [24] indicated that the concept of transformational leadership can be described through six basic behaviors: defining and articulating a vision; providing a suitable model; promoting group goal acceptance; developing high-performance expectations; providing individual support; and providing intellectual stimulation to employees. Based on this study, Carless, Wearing [25] indicated that transformational leaders articulate a vision, develop employees, provide support, empower employees, innovate ideas, act as role models, and have a charismatic style.

Transformational leaders are considered to be reliable, realistic, and effective leaders, which may enable them in achieving their goals, and they may also promote innovative work behavior [26,27]. Innovative work behavior was defined as activities that are related to an employee's development, promotion, and adoption of useful innovation at any organizational level [28]. Innovative work behavior involves the creation of new ideas, technology, and methods, as well as the trial and utilization of new techniques that are related to business processes, in particular, work aspects [15]. The process of innovation comprises both the generation and implementation of ideas [29]. As a result, it requires a wide range of unique behaviors from individuals [29].

Transformational leaders are positively associated with enhancing organizational innovation [30]. Qu, Janssen [31] conducted a study of 420 leader–follower pairs from a Chinese energy provider, and they discovered that transformational leadership has a positive impact on employees' innovative performance. Another empirical study, conducted by [15], with a sample size of 338 employees and their supervisors from 35 service and manufacturing organizations, found that transformational leadership had a positive influence on employees' innovative work behavior [15]. Based on the above arguments, the following is hypothesized:

**Hypothesis (H1).** *Transformational leadership is positively associated with innovative work behavior.*

### 2.2. Linking Transformational Leadership and Thriving at Work

Spreitzer, Sutcliffe [4] provide a socially embedded model of thriving built on the concept that its two components, learning and vitality, are strongly embedded in social systems Thriving is defined as a psychological state when individuals feel both alive and like they are learning at work [4]. The first component of thriving, vitality, is defined by Spreitzer, Sutcliffe [4] as the positive sense of having energy and a sense of being "alive". The second component, learning, refers to employees' perceptions of obtaining and using valuable knowledge and skills. According to Spreitzer, Sutcliffe [4], thriving is a personal experience that makes employees better assess their work, such as what they are doing, how they are doing it, and how they might improve it. Porath, Spreitzer [32] argued that thriving differs across work and non-work contexts, as well as responding to changes in the work environment, and that thriving was related to burnout, health, job performance, and career growth.

While the fundamental assumption of thriving at work is that high levels of vitality and learning are essential for employees to thrive [4]. Transformational leaders enhance employees' experiences of feeling "alive" and vital at work by acting as role models and motivating followers with exciting visions [11]. Additionally, transformational leaders can improve employees' learning experiences by working on supporting them in adopting a proactive learning environment, which promotes their learning ambitions and desire for development [33]. Lin, Xian, Li [33] conducted a study of 542 ordinary employees, grassroots medium, and senior managers from China and concluded that transformational leadership is significantly and positively associated with employees thriving at work.

Meanwhile, transformational leadership involves aspects that encourage and challenge employees to improve themselves such as developing standard goals, enhancing acceptance and positivity, innovating ideas, supporting, training, and influencing thriving at work [34]. Thus, the following is hypothesized:

**Hypothesis (H2).** *Transformational leadership is positively associated with thriving at work.*

*2.3. Thriving at Work as a Mediator*

Regarding the impact of thriving at work as a mediator, transformational leadership might have a positive effect on innovative work behavior for employees [15,31]. This research expects this impact through the influence of thriving at work as a mediator. From one hand, the researcher in this study claims that there is a positive impact of transformational leadership style on thriving at work. From another hand, employees' thriving at work is expected to have an influence on innovative work behavior for employees. Thriving at work serves as a criterion for monitoring employees' growth and assists in improving workplace effectiveness and flexibility [4]. An employee's innovative behavior is associated with their learning and development at the workplace, which allows them to identify problems and create solutions [35,36]. Thus, an employee needs to obtain the necessary knowledge and skills in order to comprehend the process, identify problems, and create an innovative solution [36]. In addition, employees must be willing to invest their time in adopting the new process, while social and psychological factors enable employees to thrive and be more innovative [35]. Employees who thrive more are more likely to participate in innovative behavior [10,36,37]. The pieces of evidence from previous empirical studies confirm the relationship between thriving at work and employees' innovative behavior [10,36,38]. Thus, the following is hypothesized:

**Hypothesis (H3).** *Thriving at work is positively associated with innovative work behavior.*

**Hypothesis (H4).** *Thriving at work mediates the relationship between transformational leadership and innovative work behavior.*

Based on the above discussion, Figure 1 represents the study's model:

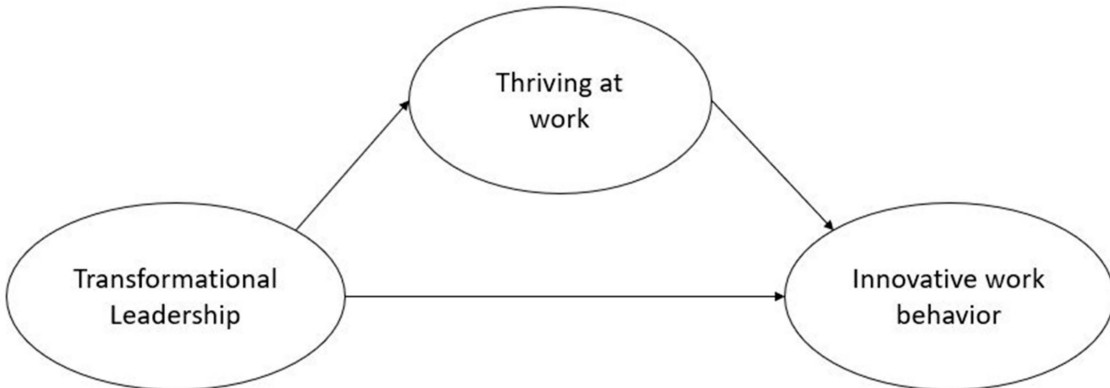

**Figure 1.** The study's model.

**3. Methods**

*3.1. Sample*

The study participants were all employees in the private sector in the Kingdom of Saudi Arabia. Samples were conveniently selected to overcome many of the issues related to research and to ensure confidentiality and anonymity [39]. The subject-to-item ratio determined the number of samples [40]. Based on this method, Bentler and Chou [40] suggested at least 5 observations per estimated parameter. Thus, the target sample size was

at least 110. The initial data sample was 289 participants. However, after cleaning the data, 224 participants who were aged 20 years and older remained as a final sample. Table 1 displays the demographic characteristics of participants. Most respondents were female (*n* = 126, 56%), unmarried (*n* = 133, 59%), and their age was between 20 and 29 years (*n* = 124, 55%). The educational level of most of the participants was a bachelor's degree (*n* = 181, 83%). Furthermore, around half of the participants had 1–5 years of experience (*n* = 102, 46%).

**Table 1.** Participants' demographic characteristics (N = 224).

| Variables | Subgroups | (N) | (%) |
|---|---|---|---|
| *Gender* | | | |
| | Female | 126 | 56 |
| | Male | 98 | 44 |
| *Nationality* | | | |
| | Saudi | 216 | 96 |
| | Non-Saudi | 8 | 4 |
| *Age* | | | |
| | 20–29 | 124 | 55 |
| | 30–39 | 69 | 31 |
| | 40–49 | 15 | 7 |
| | 50–59 | 13 | 6 |
| | 60 or more | 3 | 1 |
| *Educational level* | | | |
| | High school | 8 | 4 |
| | Diploma | 15 | 7 |
| | College degree | 187 | 83 |
| | Master's degree | 14 | 6 |
| *Marital status* | | | |
| | Married | 91 | 41 |
| | Unmarried | 133 | 59 |
| *Years of experience* | | | |
| | Less than 1 years | 42 | 19% |
| | 1–5 years | 102 | 46% |
| | 6–10 years | 35 | 15% |
| | 11–15 years | 24 | 11% |
| | More than 15 years | 21 | 9% |

*3.2. Data Collection*

This study used a self-administered online questionnaire to collect data from employees in the private sector in the Kingdom of Saudi Arabia from March to June 2022. The survey began with a brief explanation of the survey's purpose and the guidelines for completing the questionnaire. The survey included a cover letter to assure participants that their answers would remain confidential and that there are no correct or incorrect responses to limit respondents' tendency to submit biased responses [41]. The questionnaire was divided into two sections. The first section included demographic characteristics. The second section of the questionnaire was used to measure the study's variables (transformational leadership, thriving at work, and innovative work behavior). The survey was translated from English to Arabic using a forward translation by two Saudi bilingual experts who specialized in management, while the backward translation was done by a language expert who was not familiar with the management major. This language expert was asked to do back translation without having access to the initial version in order to ensure the equivalency of meaning and eliminate any meaning issues [42]. The online Google Form survey was used to collect data from the target sample.

*3.3. Measures*

3.3.1. Transformational Leadership

The global transformational leadership scale (GTL), which was established by Carless, Wearing [25], was used to assess the seven transformational leadership behaviors identified by [25]. The GTL consists of seven items (one for each behavior) that assess the frequency of the leader's transformation leadership behaviors on a 5-point Likert scale (1 = rarely or never to 5 = very frequently or always). An example of the scale items is "My leader communicates a clear and positive vision of the future". In this research, the Cronbach's alpha of this scale was 0.95.

3.3.2. Thriving at Work

A thriving at work scale, which was established by Porath, Spreitzer [32], was used to assess employees' learning and vitality with a 5-point Likert scale ("1 = strongly disagree" to "5 = strongly agree"). This scale is suitable because it assesses both learning and vitality as components of thriving. Five items were used to assess learning. An example of the items that measure learning is "At work, I find myself learning frequently". Meanwhile, five items were used to test the feeling of vitality at work. An example of the items that measure vitality is "I feel alive at work". In this research, the Cronbach's alpha of this scale was 0.90.

3.3.3. Innovative Work Behavior

Five items of scale by Scott and Bruce [29] were adapted to measure innovative work behavior with a 5-point Likert scale ("1 = strongly disagree" to "5 = strongly agree"). The scale was modified so that it could be reported by employees. An example of the scale items is "I search out new technologies, processes, techniques, and/or product idea". In this research, the Cronbach's alpha of this scale was 0.87.

*3.4. Control Variables*

Age, gender, education, and job tenure were included as control variables in this study as they have been reported in prior studies as control variables [32,43].

*3.5. Data Analysis*

The data were analyzed using Statistical Product and Service Solutions (SPSS) version 22 for descriptive purposes. To ensure the validity and reliability of the measurement model, Mplus version 8 was used to perform Confirmatory Factor Analysis (CFA) and Structural Equation Modeling (SEM).

**4. Results**

*4.1. Validity and Reliability of Measurement Models*

4.1.1. Confirmatory Factor Analysis

The measurement model was tested using Confirmatory Factor Analysis (CFA). CFA was applied to the study's variables (GTL, TAW, and IWB). Goodness-of-fit (GOF) indices were computed and compared with the GOF criteria. Based on the two-index rule, which is utilized to evaluate the goodness-of-fit of CFA, the measurement model has a good fit since CFI > 0.90 and SRMR < 0.08 [44], as shown in Table 2.

**Table 2.** GOF indices for measurement model.

| | | | | | | 90% CI for RMSEA | |
|---|---|---|---|---|---|---|---|
| $x^2$ | *Df* | **CFI** | **TLI** | **SRMR** | **RMSEA** | **LL** | **UL** |
| 3548.522 | 253 | 0.930 | 0.921 | 0.061 | 0.068 | 0.059 | 0.077 |

Notes: CFI = comparative fit index; TLI = Tucker–Lewis index; SRMR = standardized root mean square residual; RMSEA = root mean square error of approximation; CI = confidence interval; LL = lower limit; UL = upper limit.

### 4.1.2. Convergent and Discriminant Validity

The value of the average variance extracted (AVE) should be more than 0.50 to achieve convergent validity [45]. This condition was met based on the output of Table 3. In addition, composite reliability (CR) values for the three constructs were greater than 0.70. Thus, convergent validity and internal consistency were achieved in this study.

**Table 3.** Composite reliability, convergent validity, and discriminant validity.

| Construct | CR | AVE | 1 | 2 | 3 |
|---|---|---|---|---|---|
| 1-Global transformational leadership | 0.946 | 0.717 | **0.85** | | |
| 2-Thriving at work | 0.941 | 0.622 | 0.505 ** | **0.79** | |
| 3-Innovative work behavior | 0.871 | 0.532 | 0.269 ** | 0.571 ** | **0.73** |

Note(s): ** Correlation is significant at the 0.01 level (2-tailed). CR—composite reliability, AVE—average variance extracted.

In addition, since the root square values of AVE for each variable were greater than the correlations between latent variables, the research's variables had an acceptable discriminant validity, as shown in Table 3.

### 4.1.3. Common Method Bias

This study used Harman's single-factor test for common method bias to identify any bias. According to this test, if the total variance explained is less than fifty percent, the data can be analyzed because there is no common method bias. In this study, the total variance explained is 41.148, which is less than fifty percent. Consequently, there is no bias. Table 4 displays the outcome of Harman's single-factor test.

**Table 4.** Harman's Single-Factor Test.

| | Total Variance Explained | | | | | |
|---|---|---|---|---|---|---|
| | Initial Eigenvalues | | | Extraction Sums of Squared Loadings | | |
| Component | Total | % of Variance | Cumulative % | Total | % of Variance | Cumulative % |
| 1 | 9.464 | 41.148 | 41.148 | 9.464 | 41.148 | 41.148 |

### 4.2. Structural Model

### 4.2.1. Descriptive Statistics

The descriptive statistics mean, standard deviation, and correlation between the study variables are shown in Table 5. All variables have sufficient reliabilities [46]. As shown in Table 5, the value of Cronbach's alpha for global transformational leadership, thriving at work, and innovative work behavior are 0.95, 0.90, and 0.87 respectively.

**Table 5.** Mean, standard deviation, and correlation.

| | Mean | SD | Gender | Age | Edu | Exp | GTL | TAW | IWB |
|---|---|---|---|---|---|---|---|---|---|
| Gender | 0.44 | 0.497 | - | | | | | | |
| Age | 1.67 | 0.932 | −0.171 * | - | | | | | |
| Edu | 2.92 | 0.517 | −0.010 | −0.099 | - | | | | |
| Exp | 2.46 | 1.186 | −0.338 ** | 0.760 ** | −0.052 | - | | | |
| GTL | 3.0179 | 1.13161 | 0.092 | 0.092 | 0.008 | 0.053 | (0.95) | | |
| TAW | 3.4138 | 0.81622 | −0.040 | 0.083 | −0.027 | −0.001 | 0.505 ** | (0.90) | |
| IWB | 3.4122 | 0.81021 | −0.131 | 0.112 | −0.032 | 0.113 | 0.296 ** | 0.571 ** | (0.87) |

Note(s): The value of Cronbach's alpha is shown in parentheses and in italics diagonally in the matrix. * Correlation is significant at the 0.05 level (2-tailed). ** Correlation is significant at the 0.01 level (2-tailed). SD—standard deviation, Edu—educational level, Exp—years of experience in the sector, GTL—global transformational leadership, TAW—Thriving at work, IWB—Innovative Work Behavior.

### 4.2.2. Structural Model Fit

Goodness-of-fit indices were computed and compared with the GOF criteria for the structural model. Based on the two-index rule, which is utilized to evaluate the goodness-of-fit of the structural model, the structural model has a good fit since CFI > 0.90 and SRMR < 0.08 [44], as shown in Table 6.

**Table 6.** GOF indices for the structural model.

| | | | | | | 90% CI for RMSEA | |
|---|---|---|---|---|---|---|---|
| $x^2$ | *Df* | **CFI** | **TLI** | **SRMR** | **RMSEA** | **LL** | **UL** |
| 640.484 | 313 | 0.912 | 0.903 | 0.064 | 0.069 | 0.061 | 0.076 |

Notes: CFI = comparative fit index; TLI = Tucker–Lewis index; SRMR = standardized root mean square residual; RMSEA = root mean square error of approximation; CI = confidence interval; LL = lower limit; UL = upper limit.

### 4.2.3. Structural Model Results

The structural model of this study is depicted in Figure 2, which includes the control variables.

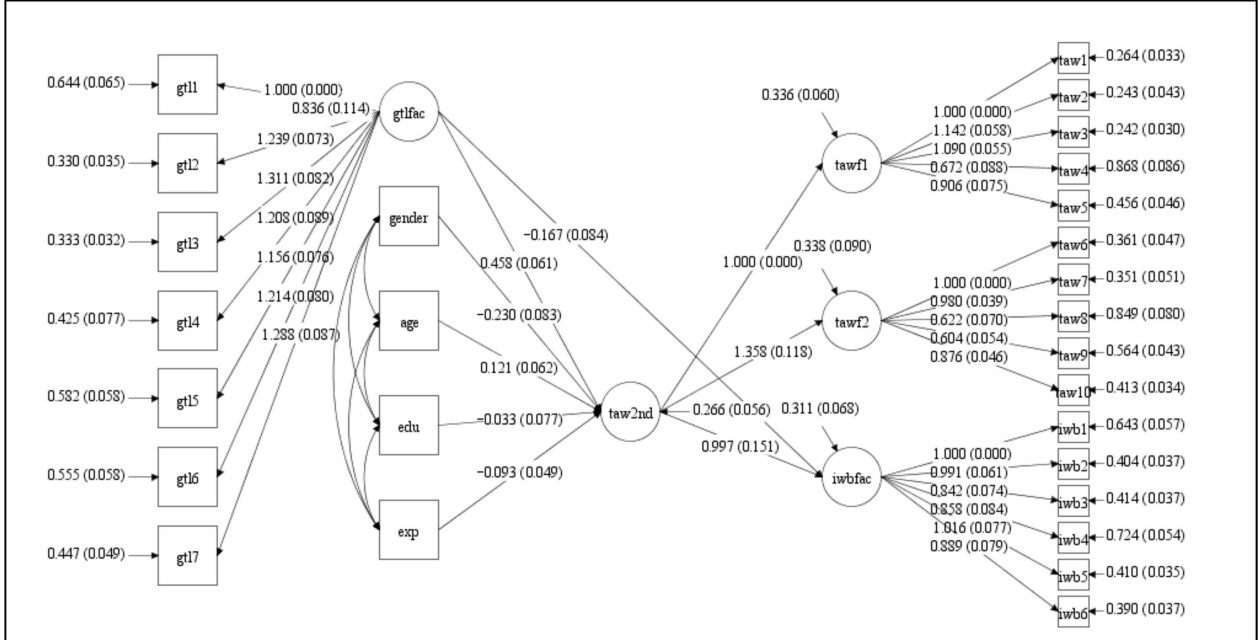

**Figure 2.** The structural model.

For the purpose of assessing the relationship between the study's variables, hypotheses were tested. As shown in Table 7, the relationship between transformational leadership and innovative work behavior was statistically significant but negative. Therefore, H1 was unsupported. The results indicate, however, that both the relationship between transformational leadership and thriving at work and the relationship between thriving at work and innovative work behavior were positive and statistically significant. H2 and H3 were thus supported. In addition, thriving at work was found to have a significant positive mediation effect on the relationship between transformational leadership and innovative work behavior. As a result, H4 was supported. Regarding the control variables, only gender has a significant impact on thriving at work.

**Table 7.** Hypotheses testing results.

| H | Relationship | Estimates | SE | Results |
|---|---|---|---|---|
| H1 | Transformational leadership → Innovative work behavior | −0.187 * | 0.095 | Not supported |
| H2 | Transformational leadership → Thriving at work | 0.619 *** | 0.055 | Supported |
| H3 | Thriving at work → Innovative work behavior | 0.829 *** | 0.094 | Supported |
| H4 | Transformational leadership → Thriving at work → Innovative work behavior | 0.513 *** | 0.088 | Supported |

Note(s): * $p$-value $\leq 0.05$, *** $p \leq 0.001$. SE—standardized error.

## 5. Discussion

Human sustainability, which can be understood as thriving at work [2], is one of the most significant phenomena to be discussed in positive organizational studies [47]. This phenomenon is significant not only in terms of how it relates to organizational outcomes and learning at work but also in terms of how it affects employees' psychological state, which has an impact on their well-being, which has become one of the most crucial issues for organizations [48]. Thus, this empirical study discussed the effect of transformational leadership on employees' thriving at work and the mediating effect of thriving at work on the relationship between transformational leadership and innovative work behavior among private sector employees in the Kingdom of Saudi Arabia. Based on the findings of this study, the impact of transformational leadership on innovative work behavior was unexpected. There was a significant but negative relationship between transformational leadership and employees' inventive work behavior. This result is opposite to most of the previous studies (e.g., [15,31]). However, a number of prior studies demonstrate that transformational leadership has a negative impact on innovative work behavior, such as [49,50]. This divergence in the study's findings could be attributed to the low perception of transformational leadership among the participants in the survey. This is in line with the findings of Bednall, E. Rafferty [50], who suggested the existence of three levels of transformational leadership (low, medium, and high) and explained that low levels of transformational leadership have a linear negative association with innovative work behavior, while high levels of transformational leadership have a positive association with innovative work behavior.

Regarding the impact of transformational leadership on thriving at work, the result shows that transformational leadership has a positive impact on thriving at work. This finding is in line with previous studies that have indicated that transformational leadership is significantly and positively associated with employees thriving at work. Lin, Xian, Li [33] conducted a study in China and concluded that transformational leadership is significantly and positively associated with employees thriving at work. This is exemplified by the fact that transformational leaders improve employees' feelings of importance at work by serving as role models and inspiring followers with inspiring visions. Furthermore, transformational leaders increase employees' learning experiences by assisting them in creating a proactive learning environment that supports their learning goals and their desire to develop.

In addition, the result shows that thriving at work has a positive impact on innovative work behavior. The present study's findings give empirical evidence for the relationship claimed by Spreitzer, Sutcliffe [4], and this result is in line with prior studies (e.g., [10,38]). These studies found that employees' feelings of thriving at work influence their innovative work behavior. This result can be explained by the fact that when employees feel vital and improve their learning experience, they are more likely to engage in innovative behavior.

Finally, our study confirmed that thriving at work mediates the relationship between transformational leadership and innovative work behavior. This result can be explained by the socially embedded model of thriving [4], which proposes that contextual factors (transformational leadership) influence thriving at work, which in turn influences positive outcomes (innovative work behavior).

## 6. Theoretical and Managerial Implications

This study has many theoretical and managerial implications. This study adds to and expands the insights in the socially embedded thriving model [4]. Spreitzer, Sutcliffe [4] recommended researchers to investigate how work unit environments, resources, and agentic work behavior influence employee thriving. This study added value to the literature on thriving at work by investigating its mediator effect on the relationship between transformational leadership and innovative work behavior. In addition, this study answers the calls to examine contextual factors that enable employees to thrive at work such as transformational leadership [14].

In practice, this study provides various paths by which leadership style influences a variety of organizational and employee outcomes. By doing so, we expect that organizations will be able to use this study to provide a supportive environment for employees in order to enhance employees' thriving at work and innovative work behavior. This research can also help managers evaluate how well their work can be designed to improve employee vitality and learning. This study, specifically, encourages organizations to establish their rules in a way that supports employees' thriving at work. The mediator impact of employees' thriving at work on the relationship between transformational leadership and innovative work behavior can benefit all the managers in the real environment by enhancing thriving at work between their employees and increasing the feeling of this concept. In addition, applying transformational leadership in the work environment will positively affect thriving at work. Thus, it is suggested to apply this kind of leadership. Also, increasing the feeling of thriving at work will make the employees more innovative. Therefore, organizations should pay attention to this concept.

## 7. The Study's Limitations and Future Research

The findings of this study are specific to the private sector in the Kingdom of Saudi Arabia. Thus, the findings cannot be generalized. As a result, it is recommended to apply this study to a large population in different sectors and contexts to generalize the findings. Furthermore, response bias often occurs when self-report is utilized to collect data. Therefore, future research could use various methods of collecting data, such as supervisor ratings for the innovative behavior variable, to generate more valid results and avoid any biases. Finally, while the result shows that gender has a significant impact on thriving at work, future studies are recommended to examine gender as a moderating variable.

## 8. Conclusions

This study concentrates on an essential topic in the management field: the effect of thriving at work as a mediator between transformational leadership and innovative work behavior. The sample of the study contained 224 employees who worked in the private sector in the Kingdom of Saudi Arabia. A quantitative approach was applied, and data were obtained using a web-based survey. The impact of transformational leadership on innovative work behavior was not supported. However, thriving at work has a mediating effect on the relationship between transformational leadership and innovative work behavior.

**Author Contributions:** Methodology, S.D.; Validation, S.D. and W.B.A.; Investigation, N.A. and S.D.; Data curation, S.D. and N.A.; Writing – original draft, N.A.; Writing – review & editing, S.D. and W.B.A.; Supervision, W.B.A. All authors have read and agreed to the published version of the manuscript.

**Funding:** The authors extend their appreciation to the Deputyship for Research and Innovation, Ministry of Education in Saudi Arabia, for funding this research work through project No. (IFKSUOR3-085-1).

**Institutional Review Board Statement:** The study was conducted in accordance with the Declaration of Helsinki, and approved by the Institutional Review Board (or Ethics Committee) of King Saud University No.: KSU-HE-22-198, Date 26/03/2022.

**Informed Consent Statement:** Informed consent was obtained from all subjects involved in the study.

**Data Availability Statement:** Data sharing is not applicable. The data are not publicly available due to participants' privacy.

**Conflicts of Interest:** The authors declare no conflict of interest.

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
