# Peer review of "Thriving at Work as a Mediator of the Relationship between Transformational Leadership and Innovative Work Behavior"

_sustainability, doi:10.3390/su151511540_

Round 1
Reviewer 1 Report
Thank you for the opportunity to read your paper. I think further exploration on the mediating effect of thriving at work between transformational leadership and innovative behaviors is strongly needed. However, when reading your paper I encountered several possibilities for improvement:
1. In the introduction section, the justification and definitions of transformation leadership and Innovative work behavior should be explained. Many studies have examined the relationship between transformational leadership and innovative behavior, so it is important to explain how your study differs from previous studies.
2. The literature cited in this study lacks recent findings. So please add relevant study findings from the last 5 years.
3. The sampling method as well as the number of questionnaires distributed and the effective return rate need to be reported.
4. How do authors detect common method bias threats?
5. In the results section, the reliability and validity tests are generally reported first with the correlation analysis.
6. The correlation analysis requires consideration of the control variables.
7. The structural equations need to report the overall model fit.
8. In the Research and Discussion section, it is recommended that the authors discuss the theoretical and practical value of this study in light of the findings, rather than just a description of the study's conclusions.
9. It is recommended that the authors rethink the Limitations and Future Research Directions of this study.
Author Response
Dear Editor,
We thank you for the opportunity to revise and resubmit. We appreciate the substantive suggestions and excellent guidance you have provided us to strengthen our contribution throughout the revision process. We address each of your points individually below. Thank you again.
- In the introduction section, the justification and definitions of transformation leadership and Innovative work behavior should be explained. Many studies have examined the relationship between transformational leadership and innovative behavior, so it is important to explain how your study differs from previous studies.
Thank you, we have added more explanations based on your comments.
- The literature cited in this study lacks recent findings. So please add relevant study findings from the last 5 years.
We have added some recent findings.
- The sampling method as well as the number of questionnaires distributed and the effective return rate need to be reported.
We have reported the missing information in regard to the sample.
- How do authors detect common method bias threats?
Harman’s single factor test is sometimes performed via a principal components analysis, and other times via an exploratory factor analysis. Harman’s single factor test entails creating a model with one single latent variable and conducting a composite-based or a factor-based analysis.
In Harman’s single factor test, the percentage of variance associated with the first component (or factor), the one referring to the highest eigenvalue, is compared against the threshold of 0.5. Such percentage is also referred to as the “total variance explained” by the first component (or factor) extracted through the analysis.
After we create a model with one single latent variable and conduct a composite based or a factor-based analysis, the “total variance explained” is in fact the average variance extracted (AVE) for the latent variable. If the AVE is greater than 0.5, in either case, one concludes that the dataset used is contaminated by common method bias.
- In the results section, the reliability and validity tests are generally reported first with the correlation analysis.
- The correlation analysis requires consideration of the control variables.
We have added them to the tables.
- The structural equations need to report the overall model fit.
We have added the model fit.
- In the Research and Discussion section, it is recommended that the authors discuss the theoretical and practical value of this study in light of the findings, rather than just a description of the study's conclusions.
- It is recommended that the authors rethink the Limitations and Future Research Directions of this study.
Thank you we have adjusted the section based on our comments.
Reviewer 2 Report
Thank you for the work on this study and the opportunity to review your manuscript. I acknowledge the importance of the survey about thriving at work. However, the following points are worthy of consideration to improve the manuscript's comprehensibility, and I hope they are helpful.
1) The manuscript needs to revise thoroughly, and paraphrasing is required. For example, the abstract is vague, with typing errors, and is unclear. Also, several sentences must be rephrased because they are copied from other works without crediting the source. For instance:
a) "The direct and indirect effects of transformational leaders."
b) "The mediating role of thriving at work in the relationship between…."
c) "Thriving at work and the direct and indirect effects of transformational leadership…."
d) "A psychological state in which individuals experience both a sense of vitality and a sense of learning at work" –I suggest paraphrasing it.
e) "The demographic characteristics of participants are shown in Table 1."
f) "A self-administered questionnaire was used to collect data in this study."
g) "translation without having access to the original text version…."
h) "finding is consistent with previous studies."
i) "impact on the relationship between transformational leadership and innovative work behavior."
2) There are so many language errors that need to be improved.
3) The study's model needs to be modified; what is the relationship between thriving at work and transformational leadership?
4) The aim of this study is not precise and needs to be improved. Also, there are no research questions.
5) The demographic table needs to be improved.
6) Data analysis followed by discussion and conclusion needs to be improved, and I wonder how the results will be.
I hope my feedback will help you to develop your research. All the best.
I suggest you need to have proofreading and edit the whole paper thoroughly, such as word choice, wordy sentences, and clarity.
Author Response
Dear Reviewer,
We thank you for the opportunity to revise and resubmit. We appreciate the substantive suggestions and excellent guidance you have provided us to strengthen our contribution throughout the revision process. We address each of your points individually below. Thank you again.
Comments and Suggestions for Authors
Thank you for the work on this study and the opportunity to review your manuscript. I acknowledge the importance of the survey about thriving at work. However, the following points are worthy of consideration to improve the manuscript's comprehensibility, and I hope they are helpful.
Thank you, we appreciate that you found much to like in our original paper and provided such excellent guidance for improving our manuscript.
1) The manuscript needs to revise thoroughly, and paraphrasing is required. For example, the abstract is vague, with typing errors, and is unclear. Also, several sentences must be rephrased because they are copied from other works without crediting the source. For instance:
- a) "The direct and indirect effects of transformational leaders."
- b) "The mediating role of thriving at work in the relationship between…."
- c) "Thriving at work and the direct and indirect effects of transformational leadership…."
- d) "A psychological state in which individuals experience both a sense of vitality and a sense of learning at work" –I suggest paraphrasing it.
- e) "The demographic characteristics of participants are shown in Table 1."
- f) "A self-administered questionnaire was used to collect data in this study."
- g) "translation without having access to the original text version…."
- h) "finding is consistent with previous studies."
- i) "impact on the relationship between transformational leadership and innovative work behavior."
2) There are so many language errors that need to be improved.
Thank you, we have now done a careful proofreading
3) The study's model needs to be modified; what is the relationship between thriving at work and transformational leadership?
We have modified the model.
4) The aim of this study is not precise and needs to be improved. Also, there are no research questions.
A research question and the aim of this study have been added.
5) The demographic table needs to be improved.
We have improved it based on your comments.
6) Data analysis followed by discussion and conclusion needs to be improved, and I wonder how the results will be.
A revision has been made in the submitted manuscript.
I hope my feedback will help you to develop your research. All the best.
Thank you for taking time to read and provide us with valuable comments

Round 2
Reviewer 1 Report
The manuscript has been revised to basically respond to my concerns and can be considered for publication in Sustainability.
Author Response
We thank you for the opportunity to revise and resubmit. We appreciate the substantive suggestions and excellent guidance you have provided us to strengthen our contribution throughout the revision process. Thank you again.
Reviewer 2 Report
Thank you for the work on this study and the opportunity to review your manuscript. I acknowledge the importance of the work. However, the following points are worthy of consideration again to improve the manuscripts’ quality, and I hope they are helpful.
1) The revised abstract still needs improvement. I wonder what the exact research method design is.
2) The method part is still vague and not clearly presented. For example, what is the difference between frequency and percentage? I suggest using variables, N, and % for the profile of participants.
3) Data analysis and results are confusing. I suggest that the authors need to present separately. For example, what are the analysis methods that the authors have used? The authors mentioned using the measurement and structural models SPSS and Mplus. Therefore, I would suggest that the authors present the data analysis method in one or two paragraphs. Then, it needs to have a result part. As for the result part, the presentation may need to revise as well. For example, in Table 7, hypothesis testing results could present as descriptive statistics and correlations through variables M, SD, and variables (1,2,3, & 4).
4) The manuscript has typing errors such as “bise’ which should be “bias,” and so forth. Therefore, the author needs to have proofreading suggested.
5) I finally wonder how long you have collected your data.
I hope my comments will help you to develop your study. All the best.
Author Response
We thank you for the opportunity to revise and resubmit. We appreciate the substantive suggestions and excellent guidance you have provided us to strengthen our contribution throughout the revision process. We address each of your points individually below. Thank you again.
Comments and Suggestions for Authors
Thank you for the work on this study and the opportunity to review your manuscript. I acknowledge the importance of the work. However, the following points are worthy of consideration again to improve the manuscripts’ quality, and I hope they are helpful.
- The revised abstract still needs improvement. I wonder what the exact research method design is.
We have added a statement in the abstract about the research method.
- The method part is still vague and not clearly presented. For example, what is the difference between frequency and percentage? I suggest using variables, N, and % for the profile of participants.
We have now used N and % as per your suggestion.
- Data analysis and results are confusing. I suggest that the authors need to present separately. For example, what are the analysis methods that the authors have used? The authors mentioned using the measurement and structural models SPSS and Mplus. Therefore, I would suggest that the authors present the data analysis method in one or two paragraphs. Then, it needs to have a result part. As for the result part, the presentation may need to revise as well. For example, in Table 7, hypothesis testing results could present as descriptive statistics and correlations through variables M, SD, and variables (1,2,3, & 4).
We have done exactly what you have suggested in the method and the results sections
- The manuscript has typing errors such as “bise’ which should be “bias,” and so forth. Therefore, the author needs to have proofreading suggested.
We have done careful proofreading.
- I finally wonder how long you have collected your data.
We have included the duration of the data collection under data collection part
I hope my comments will help you to develop your study. All the best.
Thank you for all of your comments.

Round 3
Reviewer 2 Report
Thank you for working hard on the revised one, which is great. I have one suggestion for you. As for Table 1, I suggest removing Frequency and percentage, using “N” and “%” instead. Thank you.
Author Response
Thank you. We appreciate the substantive suggestions and excellent guidance you have provided us to strengthen our contribution throughout the revision process. We have addressed our last comments and made the changes to table 1 . Thank you again.